# Climate-Adapted Potential Vegetation—A European Multiclass Model Estimating the Future Potential of Natural Vegetation

**Jonas Hinze** [1,*], **Axel Albrecht** [1] and **Hans-Gerhard Michiels** [2]

1 Department of Forest Growth, Forest Research Institute of Baden-Wuerttemberg (FVA), Wonnhaldestraße 4, 79100 Freiburg, Germany

2 Department of Forest Conservation, Forest Research Institute of Baden-Wuerttemberg (FVA), Wonnhaldestraße 4, 79100 Freiburg, Germany

* Correspondence: jonashinze@ymail.com; Tel.: +49-761-4018-766

**Abstract:** Climate change will alter the site conditions for European vegetation. This is likely to shift the potential distribution of species and habitats outside its current boundaries. To enable future projections on shifts in vegetation potentials, we fitted a multiclass model to the current potential natural vegetation (PNV) of Europe using climatic predictors. The model was then applied to climate data of the time slice 2061–2080 with the Representative Concentration Pathways (RCPs) 4.5 and RCP 8.5. With an accuracy of 0.78, simulations well represented the site-equivalent vegetation types of the current PNV across Europe. Projections show drastic shifts in vegetation potentials in all parts of Europe. Boreal forests could lose up to 75% of their current potential, while Mediterranean *Quercus* forests and steppes would double their potential area. Deserts are projected to be on the rice, and the potential of currently widespread vegetation such as *Fagus* forests would be translocated. These estimated alterations of European vegetation potentials could have great effects on the stability of current forests, affecting nature conservation strategies and forest management.

**Keywords:** climate change; random forest classification; vegetation model

## 1. Introduction

Within the coming decades, climate change will strongly affect the site conditions for European vegetation. These changing conditions will likely shift the potential distribution of species and habitats outside their current boundaries. To be able to implement measures capable of keeping pace with human-induced climate change, modeling of the magnitude and direction of expected vegetation and habitat changes is an essential first step [1].

A concept to describe the natural vegetation that would most likely occur in a certain area without human intervention is potential natural vegetation (PNV). PNV is a hypothetical concept assuming the absence of direct human impacts, such as mowing, plowing, planting and fertilizing and is especially helpful to evaluate areas without virgin vegetation. It supposes a hypothetical natural reference status of vegetation without degradation and/or unusual ecological disturbance [2,3]. Within this concept, the potential natural vegetation is the most suited species composition that will dominate in a location, given a particular set of environmental constraints. Comparing PNV to actual vegetation is a suitable tool to estimate land degradation and show vegetation potentials [2]. In its original definition, the PNV concept is static, and therefore, it does not take climate change into account [4].

PNV is used in European-managed forests as a general principle to implement the close-to-nature practice [5]. Even though flaws of the static definition are under discussion, PNV is still commonly used to assess the naturalness of existing forests [6,7]. Furthermore, PNV is applied in conservation management and land restoration to define the desired vegetation [8]. A study observing 17 habitats with different stages of succession over

25 years showed that the species composition tended to develop toward the corresponding potential natural vegetation [9], indicating a certain degree of coherence in this case.

For over two decades, a multinational group of European national experts created a map of the natural vegetation of Europe [10]. This is the first and to date only map of the potential natural vegetation for all of Europe. In a study based on this map, Hickler and Vohland et al. (2012) [11] used a generalized dynamic vegetation model to estimate the potential impact of climate change on twelve main vegetation types. Their simulations showed considerable shifts in the vegetation potentials in most areas of Europe. Using two different general circulation models (GCMs), the model projected a change in their respective potential vegetation by the year 2085 on 31%–80% of the total land area of Europe. In another simulation with an altering climate, the vegetation classes were predicted to shift their potential distribution northeast, either extending or shrinking [12]. A study modeling the vulnerability of ecosystems to vegetation shifts on a global scale found temperate mixed forest, boreal conifer and tundra and alpine biomes as the most vulnerable [13]. Substantial changes are thus expected for both current and potential vegetation as a function of global change.

In our study, we use current climatic parameters to fit a multiclass habitat suitability model (HSM). With 29 vegetation classes, the simulations have enough detailed explanatory power to be evaluated at a national or even regional level. The model was then used for simulations with future climate projections. We hypothesize that most of Europe's land surface area would be estimated to change in vegetation potentials due to altering climate. This would marginalize the meaning of today's PNV as a guideline for future-oriented ecosystem development aiming at naturalness. Our objective is to create updated climate change reference maps to help the development of climate-adapted close-to-nature forests. We focused on the demonstration of the shift in vegetation potentials in Europe.

## 2. Materials and Methods

The map of the natural vegetation of Europe consists of mosaics of homogeneous growth areas [10]. In the first two hierarchies of the map, these growth areas are differentiated by climatic site factors. Edaphic conditions are used to differentiate classes in the lower hierarchies. The vegetation class for each growth area was determined by experts by means of bioindicators, edaphic and climatic conditions. The PNV classes are therefore based on climatic factors, but the class boundaries are not defined as parametric values. To estimate potential changes in the site-equivalent vegetation types of the current PNV of Europe for different climate change scenarios, we established the concept of climate-adapted potential vegetation (CaPV). CaPV is the vegetation that would have the greatest potential to develop in an area under certain climatic conditions. To do so, we applied habitat suitability modeling (HSM) to relate hypothetical species' field observations (PNV) data to environmental predictor variables. Current climatic parameters from the time slice 1979–2013 were used to fit a multiclass HSM to 29 zonal PNV classes. To examine the effect of varying atmospheric $CO_2$ concentrations for the time slice 2061–2080, we applied two different Representative Concentration Pathways (RCPs), RCP 4.5 and RCP 8.5, to project the CaPV classes. All scientific nomenclature of plant taxa was used according to Simpson (2019) [14]. The PNV classes with the exact wording as used by Bohn and Gollub et al. (2000) [10] are accessible in the appendix for comparison (Appendix A).

### 2.1. Data and Data Preparation

ArcMap version 10.8.1. was used for the processing of all geographical data. We converted the PNV map of Europe [10] into a point grid with horizontal resolution of 4000 m for all of Europe containing the PNV classes. The data set for the model fit consisted of 550,049 points. In the vegetation map of Europe [10], the highest hierarchical level corresponds to a physiognomic–ecological classification similar to that of Ellenberg (1967) [15] containing both climatically (zonal) or edaphically-based (azonal) primary formations. At the next level, the vegetation is subdivided into broad vegetation types with

dominant species or specific species combinations in the main layer (mostly a tree layer). These class-characterizing species are the most dominant vegetation in that area. For the multiclass modeling approach, we used 29 zonal classes of the second hierarchy from the vegetation map of Europe. Zonal vegetation is determined by climatic factors, while azonal vegetation is characterized by local edaphic and topographic conditions that overrule the larger-scale effects of climate. Because of their primary association with a specific landscape type (such as coast, swamp, etc.) largely insensitive to factors of climatic change, azonal vegetation classes were excluded from the model as non-informative. For practicality, we had to aggregate some of the PNV classes with very low prevalence into one broader class (Table 1). Two other vegetation classes (E classes and N classes) were excluded from the data, because they depended on regional conditions that are not transferable. Atlantic dwarf shrub heaths (E classes) occur only when very strong coastal winds are present. Oroxerophytic vegetation (N classes) is only present on shallow soils well removed from groundwater. These classes were not considered in the simulations although listed as zonal. The emphasis in our approach lies in modeling the potential distributions of the dominant species in each vegetation class. By taking competition into account the PNV map shows the potential realized niche of these class-characterizing species or genera. Other species accompanying these dominant species may occur in different PNV classes or only in small parts of a class and can therefore not be considered in our simulations. Their distribution might underlie partly different conditions as the dominant species. CaPV does not have the explanatory scope of the PNV when it comes to species composition. CaPV represents the site-equivalent vegetation types of the current PNV of Europe for the class-dominating species.

**Table 1.** Vegetation class specification (Vegetation), class designation (CaPV class), the number of observations of the class in the dataset (n) and the model performance measures calculated using test data (Sensitivity), (Specificity) and (Balanced Accuracy).

| Vegetation | CaPV Class | n | Sensitivity | Specificity | Balanced Accuracy |
|---|---|---|---|---|---|
| Polar deserts, subnival-nival vegetation of high mountains and glaciers | A | 3192 | 0.69 | 1 | 0.85 |
| Arctic tundras | B1 | 19,280 | 0.91 | 1 | 0.95 |
| Alpine vegetation | B2 | 11,194 | 0.67 | 0.99 | 0.83 |
| Eastern boreal open woodlands | C1 | 4396 | 0.59 | 1 | 0.79 |
| Western boreal and nemoral-montane *Betula* forests | C2 | 8350 | 0.58 | 0.99 | 0.79 |
| Subalpine and oro-Mediterranean vegetation | C3 | 4769 | 0.46 | 1 | 0.73 |
| Western boreal *Picea* forests | D1 | 68,887 | 0.86 | 0.98 | 0.92 |
| Eastern boreal *Pinus-Picea* and *Abies-Picea* forests | D2 | 14,678 | 0.82 | 1 | 0.91 |
| Hemiboreal *Picea* and *Abies-Picea* forests | D3 | 35,219 | 0.83 | 0.99 | 0.91 |
| Montane to altimontane, partly submontane *Abies* and *Picea* forests | D4 | 5051 | 0.44 | 0.99 | 0.71 |
| Boreal and hemiboreal *Pinus* forests (D5) + Montane to altimontane (subalpine) *Pinus* forests (D6) | D5 | 61,311 | 0.65 | 0.96 | 0.81 |
| Species-poor acidophilous *Quercus* and mixed *Quercus* forests | F1 | 27,788 | 0.67 | 0.98 | 0.82 |
| Mixed *Quercus-Fraxinus* forests | F2 | 7953 | 0.85 | 1 | 0.92 |
| Mixed *Quercus-Carpinus* forests | F3 | 35,102 | 0.71 | 0.98 | 0.85 |
| *Tilia-Q.robur* forests | F4 | 17,559 | 0.68 | 0.99 | 0.84 |
| *Fagus* and mixed *Fagus* forests (F5) + *Fagus orientalis* forests and *Carpinus-Fagus orientalis* forests (F6) | F5 | 60,531 | 0.85 | 0.98 | 0.91 |

**Table 1.** *Cont.*

| Vegetation | CaPV Class | n | Sensitivity | Specificity | Balanced Accuracy |
|---|---|---|---|---|---|
| Caucasian mixed *Carpinus-Quercus* forests | F7 | 4444 | 0.69 | 1 | 0.85 |
| Subcontinental thermophilous (mixed) *Q. robur* L. and *Q. petraea* Liebl. forests | G1 | 3931 | 0.6 | 1 | 0.8 |
| Sub-Mediterranean-subcontinental thermophilous *Q. cerris* L. and *Q. frainetto* Ten. forests | G2 | 14,453 | 0.75 | 0.99 | 0.87 |
| Sub-Mediterranean and meso-supra-Mediterranean *Q. pubescens* Willd. forests | G3 | 13,386 | 0.63 | 0.99 | 0.81 |
| Iberian supra- and meso-Mediterranean *Q. pyrenaica* Willd., *Q. faginea* Lam., *Q. faginea* subsp. *broteroi* Cout. and *Q. canariensis* Willd. forests | G4 | 5068 | 0.68 | 1 | 0.84 |
| Meso- and supra-Mediterranean, as well as relict sclerophyllous forests | J1 | 25,968 | 0.89 | 0.99 | 0.94 |
| Thermo-Mediterranean sclerophyllous forests and xerophytic scrub | J2 | 6579 | 0.82 | 1 | 0.91 |
| Subcontinental meadow steppes and steppe-like dry grassland alternating with *Q. robur* forests | L1 | 23,214 | 0.74 | 0.99 | 0.87 |
| Sub-Mediterranean-subcontinental herb-grass steppes, partly meadow steppes alternating with oak forests | L2 | 3094 | 0.79 | 1 | 0.89 |
| True steppes | M1 | 45,652 | 0.94 | 0.99 | 0.97 |
| Desert steppes | M2 | 9106 | 0.92 | 1 | 0.96 |
| Northern lowland dwarf semishrub deserts | O1 | 7596 | 0.98 | 1 | 0.99 |
| Southern lowland-colline dwarf semishrub deserts with ephemeroids | O2 | 2298 | 0.94 | 1 | 0.97 |
| All classes | Mean | 18,967 | 0.75 | 0.99 | 0.87 |

As a source for the current and projected climate, the dataset "climatologies at high resolution for the earth's land surface areas" (CHELSA) [16] was used. CHELSA provides a global climate data set in a resolution of ~1 km for various time periods and variables, including current times (time slice 1979–2013) and several future scenarios. For the predictions of the fitted model, we selected the time slice 2061–2080 (2070) for the two RCP scenarios 4.5 and 8.5. RCP 4.5 anticipates a midrange mitigation emissions scenario with a peak of emissions around 2040. RCP 8.5 is consistent with a high emissions scenario, where greenhouse gas emissions continue to rise throughout the 21st century [17]. The future climate data were derived from four general circulation models (GCMs: HadGEM2-CC, GISS-E2-R, IPSL-CM5A-LR and MPI-ESM-LR) that were selected out of 38 possible GCMs in the CHELSA data pool. Model projections highly depend on the choice of the global circulation model used. Despite the great improvements in these models, the uncertainty of the future climate remains large [18]. To balance the uncertainties of the projections a systematic selection of global circulation models by the exclusion of the least realistic is needed to create an ensemble model. Therefore, we used the advanced envelope approach [19] to select the four GCMs for the ensemble means. As climatic predictors, a selection of the so-called bioclimatic variables (Bioclim) was used (Table 2).

**Table 2.** Specification and descriptive statistics of the current (1979–2013) climatic parameters used in the multiclass CaPV model. Min = minimum value, Max = maximum value, Mean = weighted arithmetic mean and SD = standard deviation.

| Parameter | Specification | Min | Max | Mean | SD |
|---|---|---|---|---|---|
| Bioclim 01 | Annual Mean Temperature (°C) | −13.3 | 19.6 | 6.5 | 4.9 |
| Bioclim 04 | Temperature Seasonality (SD (monthly means)) | 2.6 | 13.1 | 8.4 | 2.1 |
| Bioclim 12 | Annual Precipitation (mm/year) | 142 | 3773 | 667.6 | 273.1 |
| Bioclim 15 | Precipitation Seasonality (coefficient of variation) | 5 | 104 | 29.8 | 10.3 |
| Bioclim 18 | Precipitation of Warmest Quarter (mm/quarter) | 0 | 866 | 193.3 | 79.2 |

The Bioclim variables are supposed to be more biologically relevant than the original monthly climate layers, from which the Bioclim variables are derived [20]. In Figure 1, we provide boxplots to illustrate the relationships between the climatic variables and the PNV classes. Because of the chance that regional, annual precipitation patterns shift with ongoing climate change [21], we refrained from using Bioclims in the model that refer to the "wettest" or "driest" quarter. All climatic parameters, present and future, originate from the CHELSA dataset and were resampled to match the spatial resolution of 4 km of the response variables layer. Further specifications and values of the parameters used in the model can be viewed in Table 2.

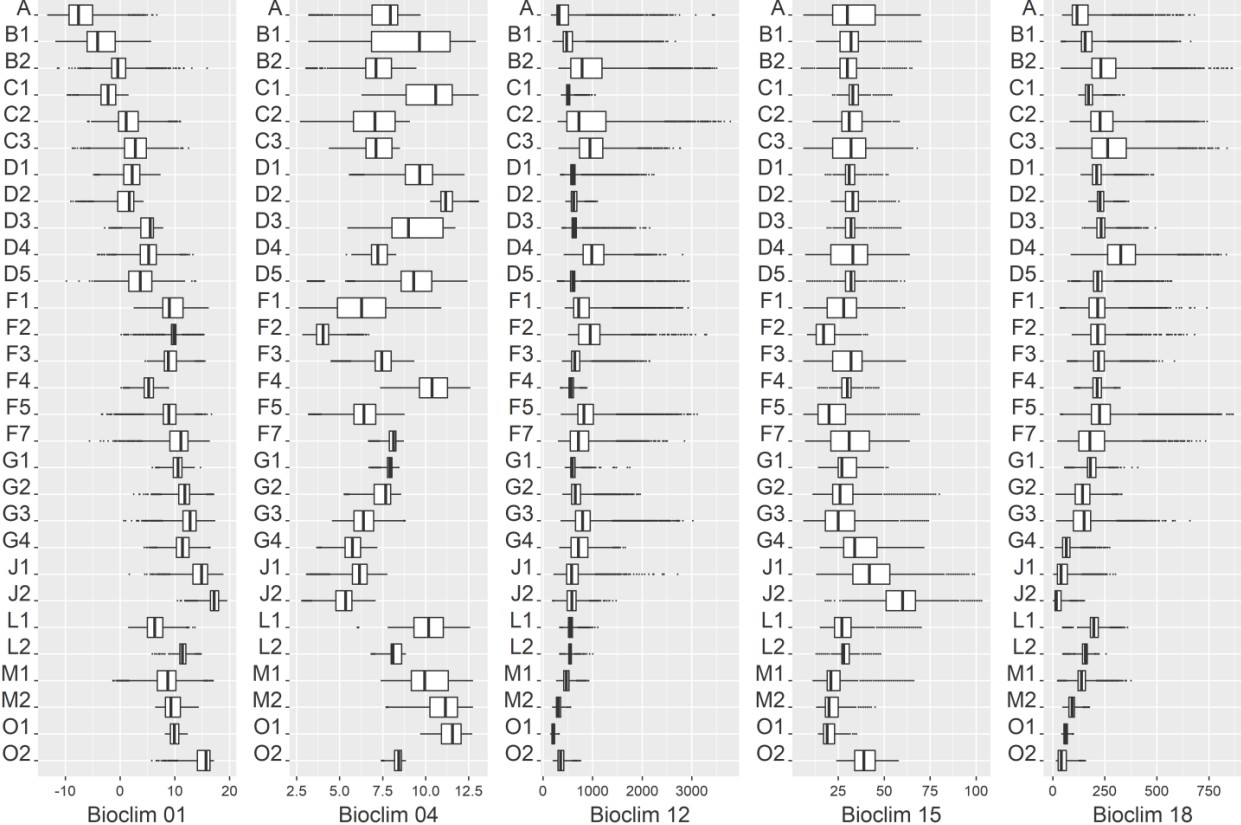

**Figure 1.** Boxplots of the climatic range of the CaPV classes within the parameters. Bioclim 01 = annual mean temperature (°C); Bioclim 04 = temperature seasonality (Std (monthly means)); Bioclim 12 = annual precipitation (mm/year); Bioclim 15 = precipitation seasonality (CV); Bioclim 18 = precipitation of warmest quarter (mm/quarter). Specifications of the CaPV classes are given in Table 1.

### 2.2. Modeling

　　　Data preparation and modeling were performed with RStudio version 1.3.959 [22]. The randomForests 4.7-1 package [23] was applied to fit a multiclass model for the 29 CaPV classes. Random forest combines several randomized decision trees, where randomness is introduced by training each decision tree on a randomly sampled subset of the training data. Random forests have been shown to efficiently handle large training datasets with many nominal classes [24]. Further advantages of this algorithm are a very high classification accuracy, its ability to model complex interactions among predictor variables, to model nonlinear and nonmonotonic relationships, to maximize information content in incomplete datasets, the capability to determine variable importance and its robustness to overfitting [25,26]. In comparison with other multiclass model approaches, random forests show a solid predicting performance, usually ranking among the top classifiers [2,27–29]. Random forests can cope with complex interactions and even highly correlated predictor variables [30]. To still avoid collinearity and keep the model as simple as possible to avoid overfitting, we selected the variables using the variance inflation factor (VIF) analysis [31]. We dismissed variables until there was no evidence of collinearity (VIF) < 5. The model was fitted with the variables Bioclims 1, 4, 12, 15 and 18. For details of the parameters see Table 2. The random forest algorithm "spreads" the variable importance across all variables. The importance of the variables (Figure 1) is explained as the mean decrease in Gini and indicates how each variable contributes to the homogeneity of the nodes and leaves in the resulting random forest. It describes the total decrease in node impurity from using a variable on the binary splits in the nodes, averaged over all trees [32,33]. If the variable is useful, it tends to split mixed-labeled nodes into pure single class nodes. With random forests, it is possible to visualize the effect of each variable for each vegetation class and make the partial dependence visible in plots. This opens up the possibility to compare the modeled effects of the variables on each class with findings in vegetation science.

　　　The random forest model was fit for the whole extent of the vegetation map of Europe with the current (time slice 1979–2013) climatic variables. The projections were made with RCP 4.5 and RCP 8.5 for the same extent with a grid resolution of 4 km. To evaluate model performance, we used the classification accuracy value (ACC), which compares the correctly classified cases with the total. The sensitivity (true-positive rate), specificity (true-negative rate) and balanced accuracy (sensitivity/2 + specificity/2) were used to show the predictive performance of each class (Table 1 and Appendix B). Cohen's Kappa can also cope with random correct classification and is therefore a "chance corrected coefficient of agreement" [34]. All of these performance indices score the performance of a model on a scale from 0.00 to 1.00, where 1.00 stands for the perfect model. In our random forest model, the optimal number of variables randomly sampled as candidates at each split (mtry) was three. Our model consisted of 400 trees having between 53,800 and 55,000 nodes each. In the 29 different classes, observations ranged from 2298 to 68,887 raster cells (Table 1) depending on their prevalence in Europe. Random forests use a different bootstrap sample for each tree. Thus, it is easy to tune without requiring an independent validation data set [32]. To still have more validation possibilities, like the calculation of Cohen's Kappa, 30% of the data (220,020 data points) were randomly left out as test data.

### 3. Results

#### 3.1. Model-Based Representation of the Current PNV Map

　　　The random forest model had an out-of-bag (OOB) error rate of 21% (see confusion matrix in Appendix B). When calculated with the test data, the model showed an overall accuracy (ACC) of 0.78, a mean balanced accuracy of 0.87 (ranging from 0.71 to 0.99) and a Cohen's Kappa value of 0.77. When comparing the original data (Figure 2A) to the corresponding model-based representation in a map (Figure 2B), very few differences can be found. Prediction errors mostly occurred as single pixels that were misclassified for a spatially neighboring class.

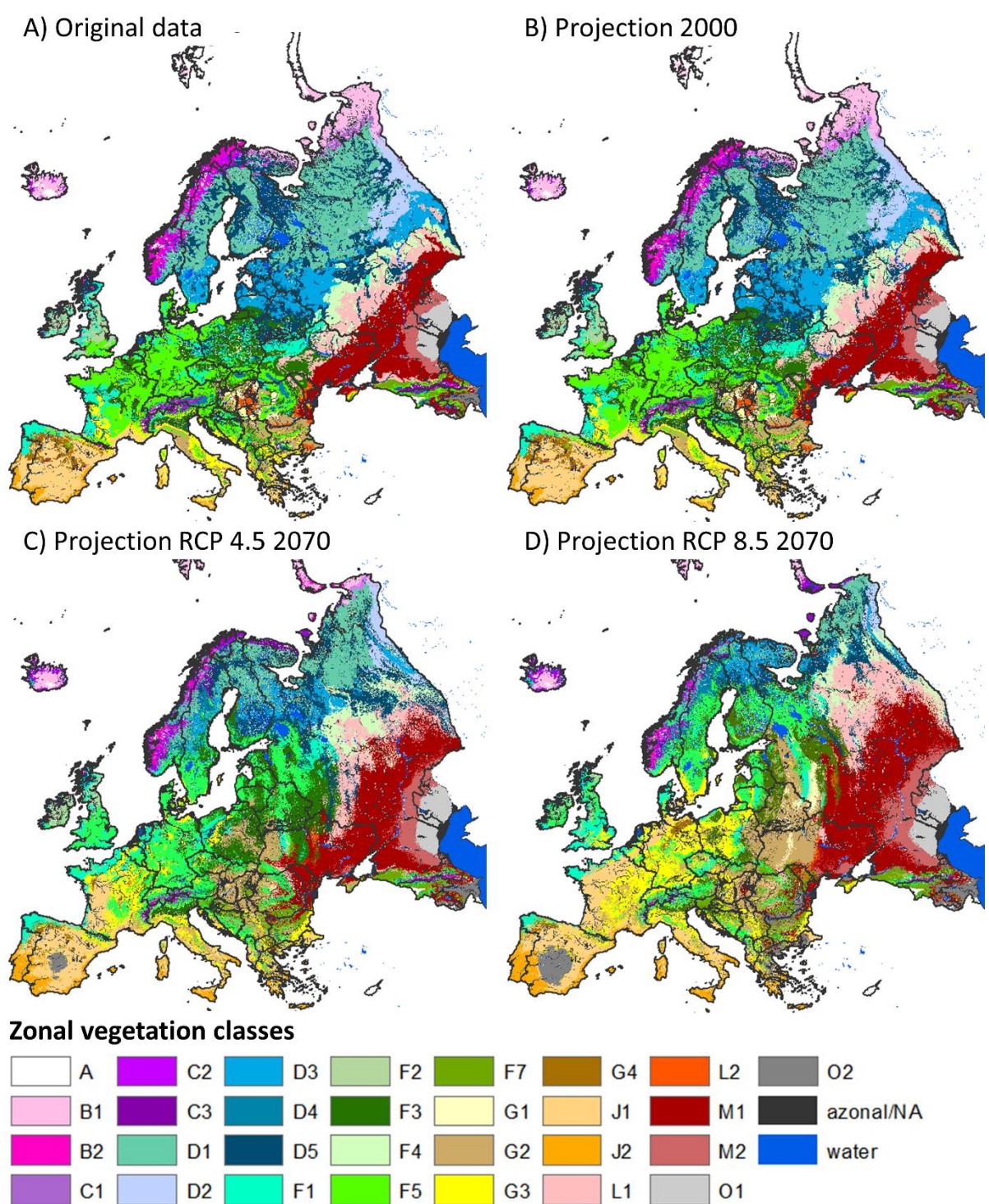

**Figure 2.** Vegetation potentials of Europe: (**A**) Original data (zonal vegetation of map of Europe (Bohn et al. 2000)), (**B**) corresponding model-based representation of this map for current climate data (averaged for 1979–2013), (**C**) projection for RCP 4.5 and (**D**) projection for RCP 8.5 for time slice 2061–2080 (2070). Class abbreviations are given in Table 1.

In the model, a low sensitivity occurred mostly in infrequent classes with a large climatic range. Classes with exceptionally low-performance measures were montane to altimontane, partly submontane *Abies* and *Picea* forests (D4), with a sensitivity of 0.44, and subalpine and oro-Mediterranean vegetation (C3), with a sensitivity of 0.46. In contrast, the predictions for arctic tundras (B1), steppes (M1 and M2) and deserts (O1 and O2)

showed outstanding performance with sensitivities above 0.9. See Table 1 for more model evaluation metrics for every vegetation class.

### 3.2. Variable Importance

Temperature Seasonality (Bioclim 04) was the most important single predictor, followed by the annual mean temperature (Bioclim 01). The precipitation of the warmest quarter (Bioclim 18) had the best predictive power among the precipitation parameters, followed by annual precipitation (Bioclim 12). The precipitation seasonality (Bioclim 15) had the least predictive importance for the model (Figure 3).

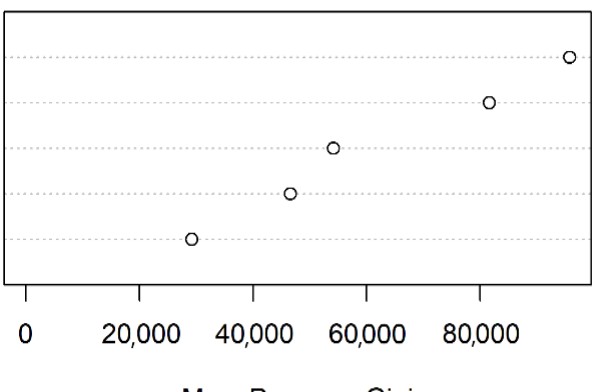

**Figure 3.** Variable importance of the fitted random forest model (mean decrease Gini) across all 29 vegetation classes.

### 3.3. Model Projections of the Climate-Adapted Potential Vegetation (CaPV)

3.3.1. Vegetation Shifts

Until 2070, our model projects drastic shifts in the potentials of European vegetation for both RCP scenarios. When comparing the two scenarios, in RCP 4.5, the vegetation potentials only partly shift outside of their current ranges, while the projections for RCP 8.5 show practically a complete spatial reorganization of the vegetation potentials (Figure 2C,D).

In northern Europe, the potential for arctic deserts (A classes) and arctic tundra (B classes) could recede, while potentials for boreal open woodlands (C) and the different types of boreal forests (D classes) would take their place (Figure 2C). In southern Scandinavia, the potentials of mixed *Quercus-Carpinus* forests (F3) and *Fagus* and mixed *Fagus* forests (F5) could replace the potential of the currently most prevalent boreal conifer forests. When using RCP 8.5 data for the prediction of the potentials of western boreal *Picea* forests (D1), eastern boreal *Pinus-Picea* and *Abies-Picea* forests (D2), hemiboreal *Picea* and *Abies-Picea* forests (D3) and boreal and hemiboreal *Pinus* forests (D5) would decline by 60%–80% until 2070 (Figure 2D).

In western Central Europe, the potentials for montane to altimontane, partly submontane *Abies* and *Picea* forests (D4), subalpine vegetation (C3) and alpine vegetation (B2) would successively ascend into higher elevations or disappear if ascent is not possible. In large parts of Central Europe, areas of *Fagus* and mixed *Fagus* forests (F5), would shift their potential toward Mediterranean *Q. pubescens* forests (G3). In eastern Central Europe, the potentials of the species-poor acidophilous *Quercus* and mixed *Quercus forests* (F1) and the mixed *Quercus-Carpinus* forests (F3) would shift their areas of potential northward. The sub-Mediterranean-subcontinental thermophilous *Q. cerris* and *Q. frainetto* forests (G2) would gradually expand their area of potential from Bulgaria and Romania to Poland, Ukraine, Belarus and Russia.

In the south and southwest of Europe, the potentials of sclerophyllous forests and xerophytic scrub (J1 and J2) would expand from its current range in manly central and southern Spain, Portugal, Sicilia, Corsica, Greece, southern France to western France, and most of

Italy, and in RCP 8.5 even to parts of Belgium, the Netherlands, northern Germany and Denmark (Figure 2D). These classes would displace mainly the potentials of Mediterranean *Q. pubescens* forests (G3), Fagus and mixed Fagus forests (F5) and species-poor acidophilous *Quercus* and mixed *Quercus* forests (F1). In the central Iberian Peninsula, climatic changes could force the transition of the potential of meso- and supra-Mediterranean, as well as relict sclerophyllous forests (J1) toward southern lowland-colline dwarf semishrub deserts with ephemeroids (O2).

The vegetation potentials in the central east of Europe are currently mainly steppes (M1 and M2) and deserts (O1 and O2). With ongoing climate change, both steppe classes could expand their potential drastically north- and westward and double their potential area by 2070 under scenario RCP 8.5.

The biggest loss on a percentage basis in both RCP projections can be seen eastern boreal open woodlands (C1), which would lose 76%–99% of its current potential, followed by polar deserts, subnival-nival vegetation of high mountains and glaciers (A), with a projected loss of 75%–97%. Even some of the very widespread classes could drastically lose potential. Western boreal *Picea* forests (D1) could lose 40%–79%, and boreal and hemiboreal *Pinus* forests + montane to altimontane (subalpine) *Pinus* forests (D5) could lose 29%–64%. Expansions are expected for Caucasian mixed *Carpinus-Quercus* forests (F7), Sub-Mediterranean-subcontinental thermophilous *Q. cerris* and *Q. frainetto* forests (G2) and sub-Mediterranean and meso-supra-Mediterranean *Q. pubescens* forests (G3), all possibly doubling their potential area. The southern lowland-colline dwarf semishrub deserts with ephemeroids (O2) could also extend their potential areas substantially. Our projections show a change in the potentials of zonal vegetation on 66% of the European land surface for RCP 4.5. and 82% for RCP 8.5. The area changes in all vegetation classes for the two RCP scenarios are visualized in absolute area values in Figure 4.

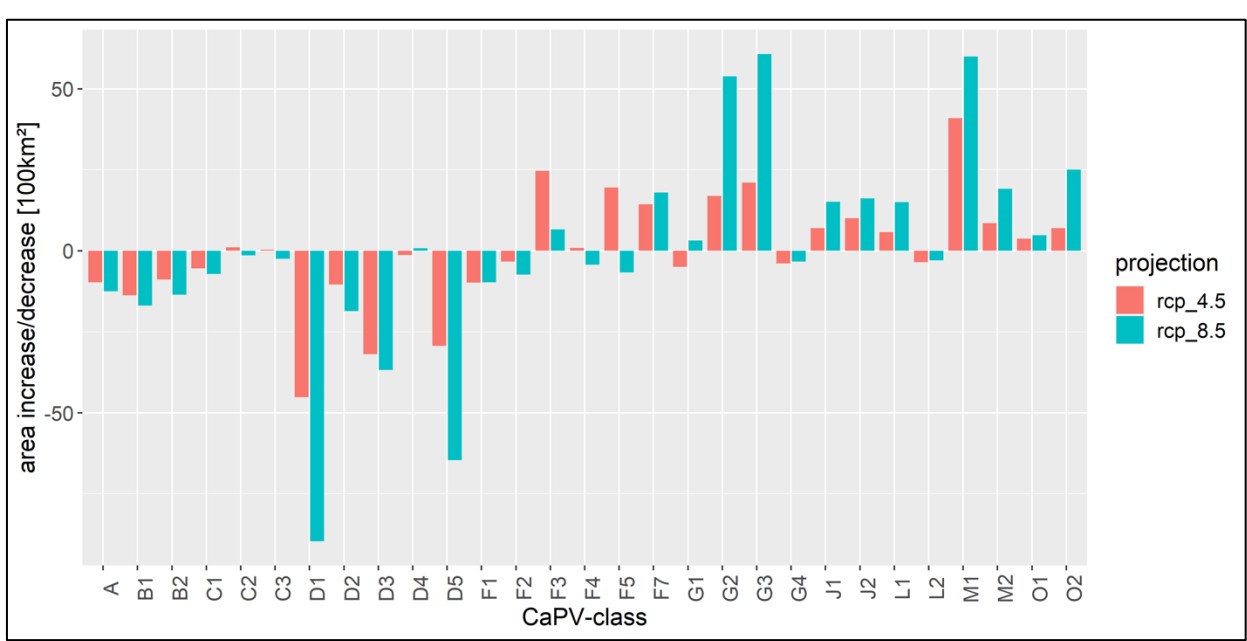

**Figure 4.** Area changes in potentials (100 km$^2$) of the climate-adapted potential vegetation classes (CaPV class) 'solid bars' until 2070 for RCP 4.5 'red' and RCP 8.5 'blue'.

3.3.2. Focus Maps

In order to illustrate the analyzability of the random forest model, we present the modeled potentials of six of the most prevalent vegetation classes of Europe in detail. We selected these six vegetation classes because of their current and future relevance for Europe. The focus maps (Figure 5) illustrate two prevalent boreal vegetation classes (D1 and D5) losing most of their potential to the *Fagus* and mixed *Fagus* forests (F5) and true steppes

(M1) with ongoing climate change. Furthermore, the potential for sub-Mediterranean *Quercus* forests (G2 and G3) could replace the potential for mixed *Quercus-Carpinus* forests (F3) and the *Fagus* and mixed *Fagus* forests (F5) as the dominant CaPV classes in Central Europe. The true steppes (M1) will extend their potential throughout eastern Europe into currently boreal and temperate areas.

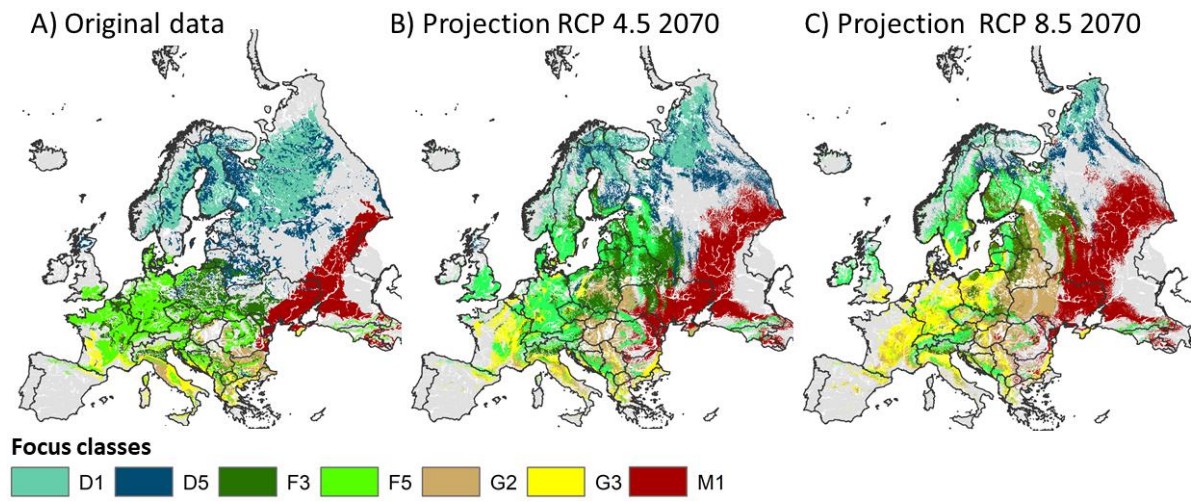

**Figure 5.** Shift in vegetation potentials of prevalent CaPV classes: Western boreal *Picea* forests (D1) 'petrol areas'; boreal and hemiboreal *Pinus* forests (D5) 'dark-blue areas'; mixed *Quercus-Carpinus* forests (F3) 'dark-green areas'; *Fagus* and mixed *Fagus* forests (F5) 'bright-green areas'; sub-Mediterranean-subcontinental thermophilous *Q. cerris* and *Q. frainetto* forests (G2) 'light-brown areas'; sub-Mediterranean and meso-supra-Mediterranean *Q. pubescens* forests (G3) 'yellow areas' and true steppes (M1) 'dark-red areas'; (**A**) Original data (zonal vegetation of map of Europe (Bohn et al. 2000)); (**B**) projection for RCP 4.5 and (**C**) projection for RCP 8.5 for the time slice 2061–2080 (2070).

3.3.3. Partial Dependence Plots

Partial dependence plots show the response curve of the class along a given predictor variable. They can be used to compare the influence of the predictors on the different vegetation classes in the model (Figure 6). Within the effect curves, the temperature seasonality (Bioclim 04) shows a high predictive power across all depicted classes. In this parameter, the curves of the classes show a high fluctuation across the full spectrum. Sub-Mediterranean and meso-supra-Mediterranean *Q. pubescens* forests (G3) and *Fagus* and mixed *Fagus* forests (F5) have a high occurrence probability (frequency) when the temperature seasonality is low. The frequency of true steppes (M1) and boreal and hemiboreal *Pinus* forests (D5) increases with the rising temperature seasonality. Instead, western boreal *Picea* forests (D1), mixed *Quercus-Carpinus* forests (F3) and sub-Mediterranean-subcontinental thermophilous *Q. cerris* and *Q. frainetto* forests (G2) occur mainly with moderate temperature seasonality. The annual mean temperature (Bioclim 01) is also applicable to distinguish between the CaPV classes across the whole spectrum. The two boreal classes (western boreal *Picea* forests (D1) and boreal and hemiboreal *Pinus* forests (D5)) mainly occur in temperatures between 3 °C and 8 °C. The occurrence possibility of the *Fagus* and mixed *Fagus* forests (F5) peaks from 6°C to 16°C, while mixed *Quercus-Carpinus* forests (F3) show a more distinct peak at 8 °C to 10°C. The two thermophilous *Quercus* classes peak toward the end of the scale at 13°C (G2) and 17°C (G3). The frequency of true steppes (M1) peaks from 10 °C to the end of the scale at 20°C. The precipitation of the Warmest quarter (Bioclim 18) is suitable to differentiate between the classes in the dry spectrum while having limited predictive power above 350 mm/year. The CaPV classes associated with very little summer precipitation (<100 mm/quarter) are true steppes (M1) and sub-mediterranean-subcontinental thermophilous *Q. cerris* and *Q. frainetto* forests (G2) followed by sub-Mediterranean and meso-supra-Mediterranean *Q. pubescens* forests (G3) (~150 mm/quarter) and western bo-

real *Picea* forests (D1) (200 mm/quarter). The classes boreal and hemiboreal *Pinus* forests (D5) and mixed *Quercus-Carpinus* forests (F3) both peak at 250 mm/quarter, while *Fagus* and mixed *Fagus* forests (F5) are associated with precipitation >400 mm/quarter. The annual precipitation (Bioclim 12) also has a limited capability as a predictor in the upper part of the spectrum (>1500 mm). In lower spectrum of this parameter there is a detailed differentiation between the classes. The true steppes (M1) have the highest frequency at 200 mm/year, followed by boreal and hemiboreal *Pinus* forests (D5) and mixed *Quercus-Carpinus* forests (F3) peaking at 600 mm/year. western boreal *Picea* forests (D1) and sub-Mediterranean-subcontinental thermophilous *Q. cerris* and *Q. frainetto* forests (G2) both have the highest frequency at 700 mm/year, followed by *Fagus* and mixed *Fagus* forests (F5) (900 mm/year) and sub-Mediterranean and meso-supra-Mediterranean *Q. pubescens* forests (G3) (1500 mm/year). The precipitation seasonality (Bioclim 15) also mainly allows a distinction between classes in the lower CV values. The classes western boreal *Picea* forests (D1), boreal and hemiboreal *Pinus* forests (D5), mixed *Quercus-Carpinus* forests (F3), sub-Mediterranean-subcontinental thermophilous *Q. cerris* and *Q. frainetto* forests (G2) and sub-Mediterranean and meso-supra-Mediterranean *Q. pubescens* forests (G3) all have their peak in occurrence probability at CV values between 20 and 30. The *Fagus* and mixed *Fagus* forests (F5) peak in this predictor at CV values of 55 and the true steppes (M1) at CV values of 65 (Figure 6).

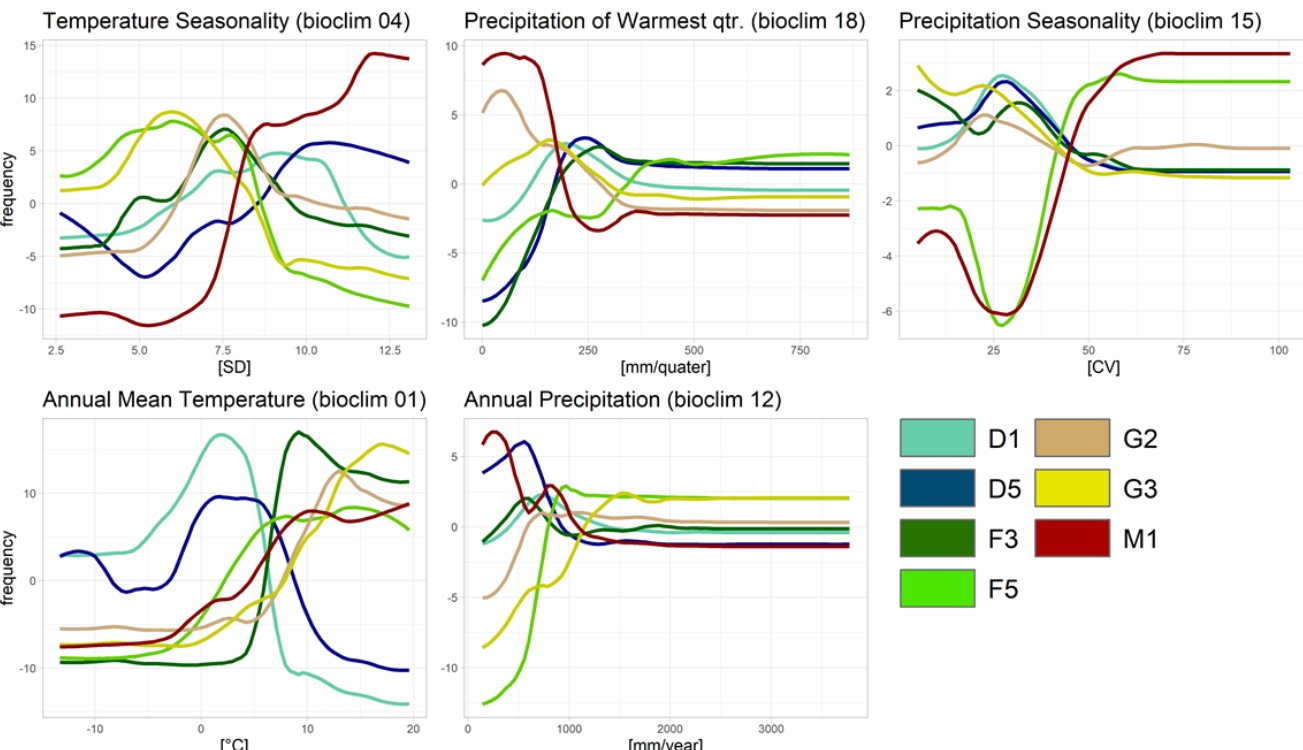

**Figure 6.** Partial dependence plots of the random forest model for prevalent CaPV classes: Western boreal *Picea* forests (D1) 'petrol lines'; boreal and hemiboreal *Pinus* forests (D5) 'dark-blue lines'; Mixed *Quercus-Carpinus* forests (F3) 'dark-green lines'; *Fagus* and mixed *Fagus* forests (F5) 'bright-green lines'; sub-Mediterranean-subcontinental thermophilous *Q. cerris* and *Q. frainetto* forests (G2) 'light-brown lines'; sub-Mediterranean and meso-supra-Mediterranean *Q. pubescens* forests (G3) 'yellow lines' and true steppes (M1) 'dark-red lines'.

## 4. Discussion

### 4.1. Shifts in Vegetation Potentials

Our projections suggest the possibility of drastic shifts in the vegetation potentials in Europe within the upcoming decades. Here, we want to provide a brief overview

of the comparability of our simulations with the findings of other vegetation modeling studies and species distribution model approaches. We additionally compare our results to studies using other theoretical approaches, such as dynamic vegetation modeling and other vegetation and empirical succession analyses. The direct comparison to these latter approaches is limited, since they describe the empirical occurrence of species, rather than the PNV. However, for generally comparing potential vegetation shifts all these studies yield interesting insights.

Our projected decline in the potential of the mesophytic and hygromesophytic coniferous and mixed broad-leaved-coniferous forests (D classes) correspond with the future estimates of the process-based generalized dynamic vegetation model (LPJGUESS) for Europe [11]. Falk and Hempelmann (2013) and Thurm et al. (2018) [35,36] found a drastic decrease in the distribution of *Picea abies* Karst. in Central Europe, which is one of the dominating species in the CaPV classes western boreal *Picea* forests (D1), eastern boreal *Pinus-Picea* and *Abies-Picea* forests (D2) and montane to altimontane, partly submontane *Abies* and *Picea* forests (D4). Hot summer drought is probably the most limiting factor for *Picea abies* [37,38]. This is in line with the very distinct decrease in the occurrence probability of western boreal *Picea* forests (D1) at 7 °C in the effect curve of Bioclim 01 (Figure 6). According to Woodward and Williams (1987) [39], the low winter temperatures limit the otherwise highly competitive broad-leaved trees from colonizing the high altitudes and boreal zone. This corresponds with the very abrupt drop of the occurrence probability for mixed *Quercus-Carpinus* forests (F3) and *Fagus* and mixed *Fagus* forests (F5) at low temperatures in the effect curves for Bioclim 01 (Figure 6). Rising temperatures enable *Fagus* and mixed *Fagus* forests (F5), as well as other broad-leaved-tree–dominated vegetation classes, to significantly expand northward. In the mountains across Europe, cold-adapted species are estimated to decline under continued climate change, and the more warm-adapted species could expand upslope [40]. This effect would be in line with our CaPV projections, where the vegetation potentials tend to shift upslope with intensifying climate change. The limitation for the occurrence of *Fagus* and mixed *Fagus* forests (F5) in the south is drought [41]. In our model, this limit is visible in the strong decline of the probability of the CaPV class *Fagus* and mixed *Fagus* forests (F5), when summer precipitation (Bioclim 18) falls below ~150 mm (Figure 6). *Q. pubescens* is well adapted to both short- and long-term drought [42,43] and is able to withstand low winter temperatures [44]. The partial dependence plots for sub-Mediterranean and meso-supra-Mediterranean *Q. pubescens* forests (G3) show a high occurrence probability for summer precipitation (Bioclim 18) between 0 and 200 mm (Figure 6) and a capability of withstanding low winter temperatures. Its tolerance to high temperatures (Bioclim 01 Figure 6) gives Mediterranean *Q. pubescens* forests advantages over other vegetation classes such as *Fagus* and mixed *Fagus* forests (F5), which are much more susceptible to heat and drought in a projected climate with extensive summer droughts. If the hydric deficits in the Mediterranean intensify due to decreasing annual precipitation and increasing air temperature, the thermophilous mixed deciduous broad-leaved forest (G classes) may be replaced by Mediterranean sclerophyllous forests and scrub (J classes) [45]. An increase in severe summer heat waves was found to have a negative impact on vegetation cover in the Iberian Peninsula, accelerating the degradation of scrub lands in the semiarid Mediterranean [46]. These conclusions are consistent with the projected expansion of the potential of deserts (O classes) in central Spain based on our simulations. In southeastern Europe, climate projections suggest a very well-defined trend toward aridity [47,48]. This goes along with the vast expansion of the potential of steppes (M classes) and deserts (O classes) our model projected for this area. The shifts in vegetation potentials in Figure 5 and the effect curves in Figure 6 are complementary. The expected increase in temperature and decrease in summer precipitation in an ongoing climate change clearly favor the hot and dry adapted vegetation classes sub-Mediterranean-subcontinental thermophilous *Q. cerris* and *Q. frainetto* forests (G2), sub-Mediterranean and meso-supra-Mediterranean *Q. pubescens* forests (G3) and true steppes (M1) (Figures 5 and 6). The effect curves illustrate this by the increased frequency of these classes in high

temperatures (Bioclim 01) and low summer precipitation (Bioclim 18). While the opposite is apparent for the boreal classes (Figure 6).

### 4.2. Considerations for Biodiversity and Conservation

Simulating with both RCP scenarios (RCP 4.5 and RCP 8.5), great changes in the potentials for natural vegetation seem most likely. Large-scale shifts in vegetation potentials can lead to the widespread decline of currently dominating species due to the disappearance of suitable habitats. Thresholds may be passed where even established individuals quickly die off, leading local ecosystems into disequilibrium stages due to lags in migration [49]. Conservation strategies must expand their planning further into the future, a process that also implies the protection of future patterns of biodiversity [1]. European species with a small range in distribution are often in locations that represent interglacial relict areas where cold-adapted species have been able to survive. These so-called rare climates refugia will potentially shrink disproportionately under future climate change, leading to high vulnerability for many of the species they contain [50]. This shrinkage of rare climate refugia can be observed for the potential of the montane and alpine vegetation in our CaPV maps. The projected lack of potential of the montane to altimontane, partly submontane *Abies* and *Picea* forests (D4), in Central Europe is an example of the vulnerability of these classes. With a shift in habitat properties as projected by our model, habitats of endemic species could be pushed beyond their ecological boundaries. Old-growth-dependent species may also be confronted with a great loss of potential habitat due to simulated rapid vegetation shifts [49]. Until 2080, more than 50% of the species conserved in the European Natura 2000 network could lose climatically suitable habitats in the protected areas [51]. Our findings suggest revising the static view of the state of conservation, e.g., in Natura 2000 surveys, to be able to assess conservation measures in a climatically dynamic environment. CaPV can provide insights into how and where vegetation potential shifts are to be expected. This knowledge is key to adjusting conservation management plans to meet the future demands of protected species.

### 4.3. Management Considerations

In field observations studying the development of succession, Prach and Tichý et al. (2016) [9] found that the estimated average time needed to reach the associated potential natural vegetation was about 180 years for primary successions and about 260 years for secondary successions. According to our model, these time periods would exceed the speed of climate-change-driven vegetation potential changes by far. An increase in tree mortality due to dry and warm conditions can already be observed throughout Europe [52–55]. Additionally, postglacial migration rates of European tree species have been reconstructed to range between 6 and 55 km per 100 years [56], while estimated biome shifts as a function of anthropogenic climate change might require much faster speeds of 300 to 700 km by the year 2100 [36,57]. We thus conclude that in many areas, climate change could drive current vegetation outside its climatic suitability long before projected suitable natural vegetation could replace it by means of natural dispersal and succession. In this condition, vegetation could shift in disequilibrium with climate, which means that plant communities do not contain all the species for which that climate is suitable [49]. The consequences may be delayed local extinctions and slow losses of ecosystem structural components. In this case, human intervention might be necessary to prevent the degradation of desired ecosystem services. Our maps can be used in management plans for effective conservation strategies, as well as forest management. Based on our results, it is highly recommended that the European nations adjust their national strategies for sustainable, close-to-nature forest management to the expected shift in potential vegetation [36,58]. The adaption of the forest policy appears advisable. Diverse forests consisting of indigenous but also nonindigenous species that might be probably suitable for the climate in a few decades should be considered and cultivated soon.

### 4.4. Methodological Considerations

Models are simplifications of reality created to enhance the understanding of a particular process or pattern. The focus of this study lies in the climate-induced changing potentials for vegetation. The realization of the projected shifts in these potentials was not the subject of this study. The used method did not include migration potentials or barriers. Our modeling approach only considered the dominating species of the vegetation classes as they were combined in the PNV map. Due to a changing climate, the competitive power of the different species might change within habitats. Furthermore, the emergence of new unique climatic combinations might lead to novel combinations of species not represented by any of the used vegetation classes. These effects could also not be considered with our method. Especially on a local scale, there are more site factors than the used variables can represent. These small-scale factors might lead to discrepancies between our prediction and the real potentials in the field. These effects may already be unrepresented in the vegetation map of Europe [10]. Areas where the climate changes in a way that there is no equivalent in Europe's current climate are outside the model boundaries. Yet, the model makes predictions without revealing that it predicts into untrained predictor combinations. To examine the climatic dissimilarity between Europe's current climate with the used future climate data the multivariate environmental similarity surface (MESS) analysis [59] was applied. The unrepresented areas differ just slightly from Europe's current climate and only make up 3% of RCP 4.5 and 6% of RCP 8.5 of the European land surface. In some areas of Europe, especially the central east, the projection maps seem somewhat blurred. Hypothetically, this could be due to creating ensemble means of the different GCMs for the climate scenarios. When projected changes are averaged over the entire area of Europe, the spatial variation of each GCM may become diluted [19].

The random forest algorithm was very suitable to create a robust classification model. The comparison between the original maps and the simulations for the present showed only minor differences. In some areas, the sharp borders of the original data were not reproduced by the model. This is caused by the fact that the vegetation map of Europe is divided in a mosaic-like fashion, where each polygon represents unique site conditions. Our prediction maps, on the other hand, consist of raster pixels with unique parameter values per pixel. The effect curves for the vegetation classes are interpretable and correspond with findings of vegetation science. The model is a novelty in the resolution of 29 vegetation classes used. Given the differentiated separation, the CaPV classes have enough explanatory power to be considered in national or even regional forest management plans.

### 5. Conclusions

With the comprehensively analyzed database, it was possible to create a robust model with 29 vegetation classes. The model displayed a very good representation of the site-equivalent vegetation types of the current PNV across Europe. Our attempt can be considered as a good trade-off between the precision of vegetation potential maps even at regional scale and generality by keeping the model as simple as possible with five predictors. With 29 vegetation classes, our approach is a novelty in precision and can therefore be seen as an innovation for further studies. The results show drastic shifts in the potentials for vegetation across Europe for both climate scenarios (RCP 4.5 and RCP 8.5). This stresses the importance of further complementary research on the potentials of additional species to obtain a more tangible understanding of vegetation able to cope with expected future conditions. This study's results may serve as a basis for forest conservation and management under global warming.

**Author Contributions:** Conceptualization A.A. and H.-G.M.; methodology, J.H., A.A. and H.-G.M.; software, J.H. and A.A.; model build, J.H. and A.A.; validation, J.H., A.A. and H.-G.M.; formal analysis, J.H. and A.A.; investigation, J.H. and A.A.; resources, A.A.; data curation, J.H.; writing—original draft preparation, J.H., A.A. and H.-G.M.; writing—review and editing, J.H., A.A. and H.-G.M.; visualization, J.H.; supervision, A.A.; project administration, A.A.; funding acquisition, A.A. All authors have read and agreed to the published version of the manuscript.

**Funding:** This research was funded in the grant program "Notfallplan" under project number 1701 by the Ministry of Food, Rural Affairs and Consumer Protection of Baden-Wüerttemberg (MLR).

**Data Availability Statement:** The datasets generated during and/or analyzed during the current study are available from the corresponding authors Jonas Hinze (jonas.hinze@forst.bwl.de) and Axel Albrecht (axel.albrecht@forst.bwl.de) on reasonable request.

**Acknowledgments:** Special thanks to Ulrich Kohnle who supervised the project and generously provided knowledge and expertise. We are also grateful to our colleagues for the very fruitful feedback sessions, and moral support.

**Conflicts of Interest:** The authors Jonas Hinze, Axel Albrecht and Hans-Gerhard Michiels declare no conflicts of interest. The funders had no role in the design of the study; in the collection, analyses, or interpretation of data; in the writing of the manuscript; or in the decision to publish the results.

**Appendix A**

List of Zonal PNV (CaPV) Classes. Nomenclature follows the vegetation map of Europe9 [10].

- A.1 = Arctic polar deserts;
- A.2 = Subnival-nival vegetation of high mountains in the boreal and nemoral zone;
- B.1 = Arctic tundras;
- B.2 = Alpine vegetation (Alpine grasslands, low creeping shrub, dwarf shrub and shrub vegetation) in the boreal, nemoral and Mediterranean zone;
- C.1 = Eastern boreal open woodlands (*Betula pubescens* subsp. *czerepanovii*, *Picea obovate*, *Pinus sylvestris*);
- C.2 = Western boreal and nemoral-montane birch forests (*Betula pubescens* s. l.), partly with pine forests (*Pinus sylvestris*);
- C.3 = Subalpine and oro-Mediterranean vegetation (forests, shrub and dwarf shrub communities in combination with grasslands and tall-forb communities);
- D.1 = Western boreal spruce forests (*Picea abies*, *P. obovate*, *P. abies* x *P. obovate*) partly with *Pinus sylvestris*, locally with birch (*Betula pubescens* s. l., *B. pendula*), alder (*Alnus incana*) or mixed forests;
- D.2 = Eastern boreal pine-spruce (*Picea obovate*, *Pinus sibirica*) and fir-spruce forests (*Picea obovate*, *Abies sibirica*), partly with *Betula pubescens* subsp. *czerepanovii*, *Larix sibirica*;
- D.3 = Hemiboreal spruce (*Picea abies*, *P. abies*, *P. obovate*, *P. obovate*) and fir-spruce forests (*Picea obovate*, *P. abies* x *P. obovate*, *Abies sibirica*) with broad-leaved trees (*Quercus robur*, *Tilia condata*, *Ulmus glabra*; *Acer platanoides*, etc.);
- D.4 = Montane to altimontane, partly submontane fir (*Abies alba*, *A. nordmannia*) and spruce forests (*Picea abies*, *P. omorika*, *P. orientalis*) in the nemoral zone;
- D.5 = Boreal and hemiboreal pine forests (*Pinus sylvestris*), partly with *Betula pubescens* s. l., *Picea obovara*, *P. abies*;
- D.6 = Montane to altimontane (subalpine) pine forests (*Pinus peuce*, *P. sylvestris*, *P. kochiana*) in the nemoral zone;
- E = Atlantic dwarf shrub heaths;
- F.1 = Species-poor acidophilous oak and mixed oak forests (*Quercus robur*, *Q. petraea*, *Q. pyranaica*, *Pinus sylvestris*, *Betula pendula*, *B. pubescens*, *B. pubescens* subsp. *Celtiberica*, *Castanea sativa*);
- F.2 = Mixed-oak–ash forests (*Fraxinus excelsior*, *Quercus robur*, *Ulmus glabra*, *Quercus petraea*);
- F.3 = Mixed-oak–hornbeam forests (*Carpinus betulus*, *Quercus robur*, *Q. petraea*, *Tilia cordata*);
- F.4 = Lime–pedunculate oak forests (*Quercus robur*, *Tilia cordata*, partly *Acer platanoides*, *A. campestre*, *Ulmus glabra*);
- F.5 = Beech and mixed beech forests (*Fagus sylvatica*, partly *F. sylvatica* subs. *Moesiaca*, *Abies alba*);
- F.6 = Oriental beech forests and hornbeam- Oriental beech forests (*Fagus sylvatica* subsp. *Orientalis*, *Carpinus betulus*);

- F.7 = Caucasian mixed hornbeam-oak forests (*Quercus robur*, *Q. petraea*, *Q. iberica*, *Q. pedunculiflora*, *Q. macranthera*, *Carpinus betulus*, *C. orientalis*, etc.);
- G.1 = Subcontinental themophilous (mixed) pedunculate oak and sessile oak forests (*Quercus robur*, *Q. petraea*, *Q. dalechampii*, *Q. polycarpa*, *Pinus sylvestris*, *Acer tataricum*);
- G.2 = Sub-Mediterranean-subcontinental themophilous bitter oak and Balkan oak forests (*Quercus cerris*, *Q. petraea*, *Q. frainetto*, *Q. dalechampii*, *Q. pedunculiflora*, *Q. pubescens*, *Q. virgiliana*, *Q. polycarpa*, *Q. hartwissiana*, *Carpinus orientalis*, *Fraxinus ornus*);
- G.3 = Sub-Mediterranean and meso-supra-Mediterranean downy oak forests, as well as mixed forests (*Quercus pubescens*, *Q. virgiliana*, *Q. trojana*, *Fraxinus ornus*, *Ostrya carpinifolia*, *Carpinus orientalis*);
- G.4 = Iberian supra- and meso-Mediterranean *Quercus pyrenaica*, *Q. faginea*, *Q. faginea* subsp. *broteroi* and *Q. canariensis* forests;
- Gla = Glaceirs;
- H = Hygro-thermophilous mixed deciduous broad-leaved trees;
- J.1 = Meso- and supra-Mediterranean, as well as relict sclerophyllous forests (*Quercus ilex*, *Q. ilex* subsp. *Rotundifolia*, *Q. coccifera*, *Q. suber*, *Pistacia lentiscus*);
- J.2 = Thermo-Mediterranean sclerophyllous forests and xerophytic scrub (*Quercus suber*, *Q. ilex* subsp. *Rotundifolia*, *Olea europaea*, *Ceratonia silique*, *Periploca angustifolia*, *Rhamnus lycioides*);
- K.1 = Pine forests and pine woodlands (*Pinus sylvestris*, *P. nigra agg.*, *P. heldreichii*, *P. halepensis*, *P. brutia*, *P. pityusa*);
- K.2 = Meso- and supra-Mediterranean fir forests (*Abies pinsapo*, *A. cephalpnica*);
- K.3 = Juniper and cypress open woodlands and scrub (*Juniperus thurifera*, *J. excelsa*, *J. foetidissima*, *J. polycarpos*, *Cupressus sempervirens*);
- L.1 = Subcontinental meadow steppes and steppe-like dry grassland (*Festuca rupicola*, *F. valesciaca*, *Stipa tirsa*, *S. pennata*, *Poa aangustifolia*, *Agrostis vinealis*) alternating with pendunculate oak forests (*Quercus robur*);
- L.2 = Sub-Mediterranean-subcontinental herb-grass steppes, partly meadow steppes (*Festuca valesciaca*, *Stipa* spp., *Bothriochola ischaemum*, *Chrysopogon gryllus*) alternating with oak forests (*Quercus pubescens*, *Q. robur*, *Q. pendunculiflora*) with *Acer tataricum*
- M.1 = True steppes (*Stipa pennata*, *S. trisa*, *S. dasyphylla*, *S. ucrainica*, *Festuca valesiaca*, *Koeleria macrantha*);
- M.2 = Desert steppes (*Stipa lessingiana*, *S. sareptana*, *Festuca valesiaca*, *Artemisia* spp.)
- N = Oroxerophytic vegetation (thorn-cushion communities, tomillares, mountain steppes, partly scrub);
- O.1 = Northern lowland dwarf semishrub deserts;
- O.2 = Southern lowland-colline dwarf semishrub deserts with ephemeroids.

## Appendix B

Confusion matrix of the random forest model with class errors.

| Class | A | B1 | B2 | C1 | C2 | C3 | D1 | D2 | D3 | D4 | D5 | F1 | F2 | F3 | F4 | F5 | F7 | G1 | G2 | G3 | G4 | J1 | J2 | L1 | L2 | M1 | M2 | O1 | O2 | Error |
|---|---|---|---|---|---|---|---|---|---|---|---|---|---|---|---|---|---|---|---|---|---|---|---|---|---|---|---|---|---|---|
| A | 1328 | 234 | 322 | 1 | 21 | 1 | 1 | 0 | 0 | 4 | 1 | 0 | 0 | 0 | 0 | 1 | 0 | 0 | 0 | 0 | 0 | 0 | 0 | 0 | 0 | 0 | 0 | 0 | 0 | 0.31 |
| B1 | 216 | 10515 | 112 | 457 | 123 | 18 | 0 | 101 | 0 | 4 | 0 | 26 | 0 | 0 | 0 | 108 | 0 | 0 | 0 | 0 | 0 | 54 | 0 | 0 | 0 | 98 | 0 | 0 | 0 | 0.09 |
| B2 | 65 | 508 | 4447 | 41 | 879 | 464 | 53 | 0 | 4 | 99 | 26 | 3 | 1 | 0 | 0 | 108 | 24 | 0 | 4 | 0 | 54 | 19 | 0 | 0 | 0 | 98 | 0 | 0 | 0 | 0.33 |
| C1 | 0 | 112 | 476 | 1580 | 30 | 350 | 224 | 0 | 0 | 2 | 78 | 0 | 1 | 0 | 0 | 0 | 1 | 0 | 0 | 0 | 0 | 0 | 0 | 0 | 0 | 16 | 0 | 0 | 0 | 0.39 |
| C2 | 6 | 125 | 1079 | 41 | 3031 | 10 | 261 | 102 | 4 | 2 | 178 | 3 | 1 | 0 | 0 | 108 | 1 | 0 | 4 | 1 | 0 | 0 | 0 | 0 | 0 | 98 | 0 | 0 | 0 | 0.40 |
| C3 | 2 | 18 | 50 | 457 | 30 | 1285 | 355 | 0 | 13 | 263 | 131 | 119 | 29 | 0 | 0 | 12 | 24 | 0 | 3 | 1 | 54 | 19 | 0 | 0 | 0 | 14 | 0 | 0 | 0 | 0.55 |
| D1 | 0 | 11 | 130 | 143 | 350 | 10 | 35814 | 483 | 378 | 2 | 562 | 0 | 2 | 0 | 0 | 37 | 0 | 4 | 6 | 7 | 0 | 5 | 3 | 71 | 0 | 169 | 0 | 0 | 25 | 0.13 |
| D2 | 0 | 0 | 1 | 1 | 8 | 0 | 430 | 7287 | 415 | 1 | 1782 | 1 | 0 | 34 | 500 | 0 | 0 | 0 | 5 | 0 | 0 | 0 | 0 | 0 | 0 | 0 | 0 | 0 | 0 | 0.17 |
| D3 | 0 | 0 | 0 | 37 | 0 | 0 | 427 | 481 | 17729 | 0 | 22 | 0 | 0 | 48 | 11 | 0 | 0 | 25 | 35 | 10 | 7 | 0 | 0 | 0 | 2 | 59 | 0 | 0 | 0 | 0.17 |
| D4 | 0 | 0 | 0 | 0 | 0 | 1 | 43 | 0 | 415 | 1323 | 24 | 1 | 2 | 0 | 10 | 23 | 19 | 4 | 6 | 7 | 0 | 0 | 0 | 5 | 2 | 187 | 3 | 0 | 4 | 0.56 |
| D5 | 0 | 0 | 0 | 0 | 0 | 67 | 6124 | 770 | 2563 | 21 | 23803 | 5 | 0 | 621 | 923 | 208 | 0 | 0 | 0 | 0 | 0 | 0 | 0 | 42 | 46 | 33 | 1 | 0 | 0 | 0.35 |
| F1 | 0 | 1 | 138 | 0 | 223 | 221 | 3 | 0 | 180 | 1323 | 553 | 11220 | 672 | 1847 | 143 | 1119 | 0 | 0 | 661 | 174 | 84 | 15 | 3 | 551 | 2 | 59 | 0 | 0 | 0 | 0.33 |
| F2 | 0 | 0 | 0 | 0 | 7 | 1 | 5 | 0 | 22 | 1 | 2 | 636 | 4077 | 0 | 11 | 37 | 19 | 0 | 35 | 10 | 0 | 0 | 0 | 71 | 0 | 76 | 0 | 0 | 0 | 0.15 |
| F3 | 0 | 0 | 15 | 0 | 43 | 2 | 2 | 0 | 180 | 4 | 2 | 78 | 0 | 15056 | 143 | 175 | 0 | 25 | 6 | 7 | 0 | 0 | 0 | 26 | 64 | 2 | 0 | 0 | 0 | 0.28 |
| F4 | 0 | 0 | 0 | 0 | 0 | 0 | 14 | 15 | 65 | 6 | 526 | 94 | 0 | 3 | 7142 | 2415 | 7 | 135 | 625 | 440 | 46 | 43 | 6 | 42 | 46 | 30 | 0 | 0 | 0 | 0.15 |
| F5 | 0 | 0 | 0 | 0 | 14 | 0 | 54 | 0 | 79 | 538 | 738 | 867 | 189 | 1694 | 0 | 30970 | 253 | 50 | 70 | 22 | 0 | 234 | 16 | 14 | 88 | 33 | 0 | 0 | 4 | 0.35 |
| F7 | 0 | 1 | 53 | 0 | 27 | 7 | 14 | 15 | 0 | 0 | 94 | 5 | 0 | 52 | 0 | 335 | 1814 | 11 | 398 | 9 | 0 | 0 | 6 | 5 | 46 | 0 | 0 | 0 | 0 | 0.32 |
| G1 | 0 | 0 | 0 | 0 | 0 | 0 | 2 | 0 | 2 | 12 | 3 | 51 | 0 | 326 | 0 | 105 | 9 | 1413 | 869 | 22 | 0 | 0 | 0 | 0 | 7 | 0 | 0 | 0 | 0 | 0.41 |
| G2 | 0 | 0 | 5 | 0 | 0 | 27 | 0 | 0 | 3 | 0 | 7 | 18 | 19 | 384 | 0 | 562 | 22 | 200 | 6488 | 35 | 4 | 0 | 0 | 0 | 0 | 0 | 0 | 0 | 0 | 0.25 |
| G3 | 0 | 0 | 1 | 0 | 6 | 7 | 0 | 0 | 0 | 0 | 0 | 96 | 2 | 226 | 0 | 838 | 32 | 45 | 869 | 5138 | 486 | 713 | 3177 | 0 | 7 | 0 | 0 | 0 | 0 | 0.36 |
| G4 | 0 | 0 | 0 | 0 | 0 | 6 | 0 | 0 | 0 | 0 | 1 | 4 | 19 | 1 | 0 | 40 | 0 | 0 | 114 | 484 | 2151 | 0 | 468 | 0 | 4 | 0 | 0 | 0 | 0 | 0.29 |
| J1 | 0 | 0 | 0 | 0 | 0 | 0 | 0 | 0 | 0 | 13 | 0 | 37 | 0 | 1 | 0 | 69 | 0 | 0 | 3 | 18 | 486 | 13986 | 16 | 0 | 0 | 0 | 0 | 0 | 0 | 0.11 |
| J2 | 0 | 0 | 0 | 0 | 0 | 0 | 0 | 0 | 0 | 0 | 0 | 4 | 0 | 1 | 0 | 0 | 0 | 0 | 0 | 0 | 4 | 713 | 3177 | 0 | 0 | 0 | 0 | 0 | 0 | 0.19 |
| L1 | 0 | 0 | 0 | 0 | 0 | 82 | 3 | 0 | 249 | 10 | 353 | 115 | 0 | 78 | 103 | 86 | 154 | 70 | 44 | 23 | 0 | 2 | 0 | 10322 | 88 | 142 | 231 | 74 | 79 | 0.23 |
| L2 | 0 | 0 | 0 | 0 | 1 | 0 | 1 | 0 | 0 | 0 | 0 | 4 | 0 | 56 | 0 | 7 | 1 | 14 | 113 | 23 | 0 | 0 | 0 | 5 | 1440 | 734 | 0 | 0 | 0 | 0.26 |
| M1 | 0 | 0 | 6 | 0 | 0 | 2 | 0 | 0 | 0 | 6 | 353 | 0 | 0 | 701 | 1244 | 59 | 44 | 14 | 51 | 18 | 0 | 2 | 0 | 571 | 88 | 25658 | 5119 | 4447 | 0 | 0.08 |
| M2 | 0 | 0 | 0 | 0 | 0 | 0 | 0 | 0 | 0 | 0 | 0 | 0 | 0 | 0 | 0 | 7 | 1 | 0 | 0 | 0 | 0 | 0 | 0 | 0 | 1440 | 340 | 5119 | 74 | 0 | 0.02 |
| O1 | 0 | 0 | 0 | 0 | 0 | 0 | 0 | 0 | 0 | 0 | 0 | 0 | 0 | 0 | 0 | 0 | 0 | 0 | 0 | 0 | 0 | 0 | 0 | 0 | 0 | 0 | 85 | 4447 | 0 | 0.02 |
| O2 | 0 | 0 | 0 | 0 | 1 | 0 | 0 | 0 | 0 | 0 | 0 | 0 | 0 | 0 | 0 | 0 | 22 | 0 | 0 | 0 | 0 | 0 | 0 | 2 | 0 | 62 | 0 | 0 | 1270 | 0.06 |

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
