# Peer review of "Climate-Adapted Potential Vegetation—A European Multiclass Model Estimating the Future Potential of Natural Vegetation"

_forests, doi:10.3390/f14020239_

Round 1
Reviewer 1 Report
I have carefully reviewed the manuscript titled “Climate-adapted potential vegetation - a European multiclass model with a case study in Baden-Wuerttemberg”.
Overall I appreciated the work and think that the research can be a meaningful contribution to the literature. However, I do have a few important methodological issues that concern me about the results from the effort.
MAIN ISSUE 1: My main concern with the work is that the authors made little effort to ensure that their models was not overfitting the training data, and hence had little power to generalize climate vegetation relationship that can then be applied in novel (future) scenarios. Indeed as can be inferred from table 3, most uncommon classes are misclassified as likely common classes (low sensitivity/ high specificity), meaning the models may have a bias towards classifying pixels as some of the most common classes considered. This makes me wander if, when applying the model into novel future conditions, the model’s bias towards common classes is partly responsible for the large shifts towards some of these large classes. Overall, I feel like this could be one situation where less is better: had the authors focused on a more aggregated set of classes, used fewer predictors, and had a more truly independent test set, the results would likely be a lot more trustworthy.
Specific overfitting, related issues:
L129: It is still concerning that so many variables were considered, which increases the possibility for model overfitting.
L170: Even if RF supposedly can handle highly collinear datasets, it still makes it very hard to make inferences about what factors really matter in shaping vegetation distribution. Also, random forest can over fir data,
especially if you have highly correlated variables and a lot of sample points that are highly spatially autocorrelated.
L225: You should not include a bunch of correlated variables and then interpret model variable importance individually as the authors themselves mention that variable importance of correlated variables tend to be spread out across them. Same concern goes for Figure 1.
L198: A RF with nearly 70 thousand nodes is huge, and could easily be overfitting your data.
L199-202: As the authors mention, there are a few papers out there about how RF is relatively immune to overfitting, but anyone that has done any spatial modeling using RF will know that it still can hugely overfit data. I am glad that the authors set aside 10% of the dataset for independent evaluation. However, because of spatial autocorrelation, a random subset of pixels will be nearly identical to surrounding pixels used for training, meaning your model evaluation is not truly independent. A lot of effort has been laid on coming up with more independent test data. For instance, r package ENMeval has a good set of functions to partition data into training and evaluation sets in more robust ways.
MAIN ISSUE 2:
The authors seem to somewhat brush aside the fact that the model performed rather poorly for a lot of classes, especially the less common ones. The clear match between modeled map and the original data can be entirely due to overfitting issues raised above. This is especially true if 90% of the original map was used for model fitting and the remaining 10% is extremely similar to the training set. Additionally, even if overfitting is not an issue, the model map may still look generally accurate at a coarse scale given that the model did ok the more common large classes.
Table 3 is a good illustration of this concern about the general quality and utility of the model for the less common classes: The sensitivity values for a lot of the less common classes are actually quite low (in some cases <20%). Given that specificity of your model was so high, even without a confusion matrix (which would have been nice as an appendix), we can see that likely a lot of the uncommon classes were being classified as the dominant cover classes, which certainly would be problematic for those classes.
The fact that your sensitivity and specificity for most of your classes was so unbalanced, resulting in a likely under representation of most classes considered, is something that should have been more significantly addressed. Indeed the models seem to work well for the dominant cover classes, but certainly it does not do well for small classes that may indeed be of conservation interest.
MODERATE ISSUES:
L313: I am not sure if the case study adds much to the paper. I would perhaps add it as an appendix as an example of how the model can possibly be applied at finer scales.
L445: Management recommendation is quite generic. Also, given that the model does quite poorly with smaller classes, I am not sure this actually makes sense! Given some of the quality issues discussed above, a more nuanced recommendation is in order.
In general, it is really awkward to have to check a look up table for each class since the class abbreviations are entirely non-descriptive. I would perhaps switch a slightly longer (but more useful) set of abbreviations.
L43: Has there been meaningful recent climatic changes that may have already shifted these PNVs? If so, how would that impact your analyses?
L92: Details about how the original PNV map was created is needed to understand if and how we should expect the model to fit the data
L94: What was the resolution of the PNV map? You should indicate here (not way down below) how much was left for model evaluation
L104: This is not clear: up to now the paper describes modeling vegetation classes, but now it switches to individual species. Also unclear how competition is included in the model.
L133: This does not make much sense- the quarterly based metrics would still give an indication of quarterly extremes
L143: I don’t understand: if you exclude azonal classes that are not defined by climatic factors, why are you still including edaphic factors?
L380: I appreciate the section in the discussion were the results are cross validated with past efforts.
L480-503: this entire paragraph is really roughly written and needs a good proof read; L487: fragment; L495-497: move to methods?
MINOR ISSUES:
L40: Don’t use climax community as this research is not about succession, but the most suited species ensembles to a particular area.
L57: Again, this research is not about succession
L50: Don’t say that vegetation has evolved.
L52: delete ‘project a’
L58: can you indicate a range of scenarios considered?
L59: on -> of
L60: delete site?
L85: what time range for scenarios considered?
L128: Add coma after predictors
L275: I would switch simulations to projections, as these are not technically simulations
L34: site -> future?
L480: solid (too informal)-> robust?
Author Response
Dear reviewer 1.
Thank you for your comprehensive review. We have carefully considered all your concerns and were able to implement the proposed changes. For us this was an insightful process that improved our manuscript substantially.
MAIN ISSUE 1: My main concern with the work is that the authors made little effort to ensure that their models was not overfitting the training data, and hence had little power to generalize climate vegetation relationship that can then be applied in novel (future) scenarios. Indeed as can be inferred from table 3, most uncommon classes are misclassified as likely common classes (low sensitivity/ high specificity), meaning the models may have a bias towards classifying pixels as some of the most common classes considered. This makes me wander if, when applying the model into novel future conditions, the model’s bias towards common classes is partly responsible for the large shifts towards some of these large classes. Overall, I feel like this could be one situation where less is better: had the authors focused on a more aggregated set of classes, used fewer predictors, and had a more truly independent test set, the results would likely be a lot more trustworthy.
RESPONSE: We considered your objections towards our model approach. Some of the vegetation classes had very few representations (<1000) for a model with ~500 000 data points and 30+ classes. We excluded some of the very small classes, or aggregated them into one equivalent class of the higher hierarchy. For example: A1 + A2 -> A. Other classes like K: Xerophytic coniferous forest vegetation that where originally listed a zonal, but had also azonal characters like “shallow soils well removed from groundwaters”. Classes that are partly azonal or have other regionally limited conditions like very strong winds (Atlantic dwarf shrub heaths) were also excluded. Hence we reduced the dataset to 29 classes with the smallest class having ~2000 representations. We also modified the variable selection process to become more rigorous as to develop more parsimonious models.
Specific overfitting, related issues:
L129: It is still concerning that so many variables were considered, which increases the possibility for model overfitting.
RESPONSE: We decided to revise the model. We reduced the variables using the variance inflation factor (VIF) from 12 to 5 predictors. This decreased the model performance slightly. The projections stayed mostly similar to the projections in the first version of the manuscript while using a much less complex model.
L170: Even if RF supposedly can handle highly collinear datasets, it still makes it very hard to make inferences about what factors really matter in shaping vegetation distribution. Also, random forest can over fir data, especially if you have highly correlated variables and a lot of sample points that are highly spatially autocorrelated.
RESPONSE: We reduced the variables only allowing a VIF of less than 5. Usually values from 5 to 10 are considered as critical for multi- variable correlation.
Variables VIF
bioclim_01 1.827728
bioclim_04 2.177615
bioclim_12 2.600584
bioclim_15 1.054777
bioclim_18 2.260466
L225: You should not include a bunch of correlated variables and then interpret model variable importance individually as the authors themselves mention that variable importance of correlated variables tend to be spread out across them. Same concern goes for Figure 1.
RESPONSE: The new model does not include correlated variables.
L198: A RF with nearly 70 thousand nodes is huge, and could easily be overfitting your data.
RESPONSE: The new model still has ~55.000 nodes. A classification tree in the random forest algorithm has a minimum size of terminal nodes of one observation. Meaning it keeps splitting the nodes until it has pure terminal nodes including only one class. In a classification tree with 29 classes that needs a lot of splits, even if the model is fitted with only five variables. So the huge size is partly inherent to multiclass RF.
L199-202: As the authors mention, there are a few papers out there about how RF is relatively immune to overfitting, but anyone that has done any spatial modeling using RF will know that it still can hugely overfit data. I am glad that the authors set aside 10% of the dataset for independent evaluation. However, because of spatial autocorrelation, a random subset of pixels will be nearly identical to surrounding pixels used for training, meaning your model evaluation is not truly independent. A lot of effort has been laid on coming up with more independent test data. For instance, r package ENMeval has a good set of functions to partition data into training and evaluation sets in more robust ways.
RESPONSE: I have read into the ENMeval description and usage and could not find a way to use this package for multiclass models. It needs occurrences and background records (or pseudo-absence) to be used. So in my understanding ENMeval can only be used for presence / absence data. Additionally, data partitioning / splitting will never create truly indipendent evaluation data, since they are always from the same study design. By definition never independent.
MAIN ISSUE 2:
The authors seem to somewhat brush aside the fact that the model performed rather poorly for a lot of classes, especially the less common ones. The clear match between modeled map and the original data can be entirely due to overfitting issues raised above. This is especially true if 90% of the original map was used for model fitting and the remaining 10% is extremely similar to the training set. Additionally, even if overfitting is not an issue, the model map may still look generally accurate at a coarse scale given that the model did ok the more common large classes.
RESPONSE: We simplified the model drastically (reclassify the rare classes, reduce number of predictors, as described above) and set aside 30% instead of only 10 % of the data as training data. So we tried to considered the objections of overfitting the model as much as possible. The projection of the current map is still very accurate, not projecting any class into areas where the class does not appear in the original map. Considering that we are using 29 classes across Europe the model is able to project the correct classes even if they are strongly mixed in the same area. If a class is very common the model does not just put the class all over the map. It still represents the relatively delicate differences in areas with many classes in the same region. Small classes can also have a very high sensitivity and are well represented in the maps.
Table 3 is a good illustration of this concern about the general quality and utility of the model for the less common classes: The sensitivity values for a lot of the less common classes are actually quite low (in some cases <20%). Given that specificity of your model was so high, even without a confusion matrix (which would have been nice as an appendix), we can see that likely a lot of the uncommon classes were being classified as the dominant cover classes, which certainly would be problematic for those classes.
RESPONSE: We included the confusion matrix in the appendix for a better transparency. We also tried to deal with the issue of misclassifying the small classes by aggregating them into an equivalent class of the higher hierarchy. In a deeper analysis of two of the classes with low sensitivity we found non zonal characteristics and excluded them. The classes with the lowest sensitivity now have values of 44% and 46%. All other classes have a sensitivity of 58 % to 98 %. I hope the model as it is can be seen as a good compromise between not to being complex (not overfitted) and still being appropriate in the representation of the classes.
The fact that your sensitivity and specificity for most of your classes was so unbalanced, resulting in a likely under representation of most classes considered, is something that should have been more significantly addressed. Indeed the models seem to work well for the dominant cover classes, but certainly it does not do well for small classes that may indeed be of conservation interest.
RESPONSE: The fact that sensitivity and specificity for most the classes is so unbalanced is explainable by the fact, that we used so many classes. The specificity with 29 classes all having a prevalence >0.12 should always be high, unless one class was immensely over predicted. We checked the correlation between the prevalence and the sensitivity of the classes in the new model. The correlation coefficient was 0.24 indicating that the relationship between a low prevalence and a low sensitivity is not so apparent.
MODERATE ISSUES:
L313: I am not sure if the case study adds much to the paper. I would perhaps add it as an appendix as an example of how the model can possibly be applied at finer scales.
RESPONSE: We entirely removed the case study
L445: Management recommendation is quite generic. Also, given that the model does quite poorly with smaller classes, I am not sure this actually makes sense! Given some of the quality issues discussed above, a more nuanced recommendation is in order.
RESPONSE: There are no more really poorly fitted small classes in the model.
In general, it is really awkward to have to check a look up table for each class since the class abbreviations are entirely non-descriptive. I would perhaps switch a slightly longer (but more useful) set of abbreviations.
RESPONSE: I could not find a more intuitive way of naming the classes. There is for example 11 classes with the taxa “Quercus” included in the name. Another 6 classes are Mediterranean classes. So I fear another way would make the names longer, but not easier to grasp.
L43: Has there been meaningful recent climatic changes that may have already shifted these PNVs? If so, how would that impact your analyses?
RESPONSE: The Map of the natural vegetation of Europe was published in 2000. The time slice for the current climate data is 1979 – 2013 (~1996). So I think the two datasets complement each other very well. Recent climatic changes have already an effect on vegetation as described in the chapter 4.1 Shifts of vegetation potentials. There we described already occurring shifts in vegetation (L371-373 & L377-379).
L92: Details about how the original PNV map was created is needed to understand if and how we should expect the model to fit the data
RESPONSE: We added text. In L75-80: “The map of the natural vegetation of Europe consists of mosaics of homogeneous growth areas [10]. In the first two hierarchies of the map these growth areas are differenti-ated by climatic site factors. Edaphic conditions are used to differentiate classes in the lower hierarchies. The vegetation class for each growth area was determined by experts by means of bio-indicators, edaphic- and climatic conditions. The PNV classes are therefore based on climatic factors but the class boundaries are not defined as parametric values. “
L94: What was the resolution of the PNV map? You should indicate here (not way down below) how much was left for model evaluation
RESPONSE: The map of the natural vegetation of Europe is a vector map. So it has no resolution.
L104: This is not clear: up to now the paper describes modeling vegetation classes, but now it switches to individual species. Also unclear how competition is included in the model.
RESPONSE: L103: “These class characterizing species are the most dominant vegetation of that area.” In the PNV map competition is included. The PNV class is always the dominant vegetation of that area.
L133: This does not make much sense- the quarterly based metrics would still give an indication of quarterly extremes
RESPONSE: As example: In central Europe the wettest quarter in the year 2000 was the summer. With ongoing climate change annual precipitation patterns shift towards the winter. So depending on the climate scenario and area bioclim 16 (mean monthly precipitation amount of the wettest quarter) could mean the summer precipitation in the year 2000 and the winter precipitation in the year 2070. To avoid this mix of information we chose bioclim 18 (mean monthly precipitation amount of the warmest quarter) because the inter annual temperature pattern is not projected to change.
L143: I don’t understand: if you exclude azonal classes that are not defined by climatic factors, why are you still including edaphic factors?
RESPONSE: We excluded the edaphic factors.
L380: I appreciate the section in the discussion were the results are cross validated with past efforts.
RESPONSE: Thank you
L480-503: this entire paragraph is really roughly written and needs a good proof read;
RESPONSE: We edited this paragraph
L487: fragment; L495-497: move to methods?
MINOR ISSUES:
L40: Don’t use climax community as this research is not about succession, but the most suited species ensembles to a particular area.
RESPONSE L38: “Within this concept the potential natural vegetation is the most suited species composition that will dominate in a location, given a particular set of environmental constraints.”
L57: Again, this research is not about succession
RESPONSE L55: “Their simulations showed considerable shifts of the vegetation potentials…”
L50: Don’t say that vegetation has evolved.
RESPONSE: L48: “A study observing 17 habitats with different stages of succession over 25 years showed that the species composition tended to develop towards the corresponding potential natural vegetation…”
L52: delete ‘project a’
RESPONSE: L51: “For over two decades a multinational group of European national experts…”
L58: can you indicate a range of scenarios considered?
RESPONSE: L57: “Using two different general circulation models (GCM), the model projected a change in their respective potential vegetation…”
L59: on -> of
RESPONSE: Done
L60: delete site?
RESPONSE: deleted “site” and “factors”
L85: what time range for scenarios considered?
RESPONSE: L88: “To examine the effect of varying atmospheric CO2-concentrations for the time slice 2061 – 2080 we applied two different Representative Concentration Pathways…”
L128: Add coma after predictors
RESPONSE: done
L275: I would switch simulations to projections, as these are not technically simulations
RESPONSE: done
L34: site -> future?
RESPONSE: Site conditions is an established term in vegetation science. Future conditions would still be meaningful, but less precise.
L480: solid (too informal)-> robust?
RESPONSE: yes, robust sounds better
Reviewer 2 Report
Dear authors,
First, I would like to thank you for presenting your results. Overall, I found the work very interesting. The study design setup and analysis were well performed. The article is understandably written and well-organized. The methods are clearly explained, the results are well described, and the discussion is carried out very well. The bibliography is good and sufficient. In my opinion, good paper. So, I think you did a very good job.
I only have a few minor suggestions for you to consider in the revision.
1. Section 2. Materials and Methods; 2.1. Data and data preparation; lines 113-116, Table 1., and in some parts in Results and Discussion.
In the model, you used climatic data on the current conditions, which you define with relatively little accuracy; e.g "current climate; variables including current times ". I guess that these are data from the reference period 1981-2010 (such data is in CHELSA). This should be precisely specified in the article. Please add in subsection 2.1 information about the current climate data. (e.g. average values from the period .......). As your article shows, the climatic input data has a great influence on the final result of the model, hence information on this is necessary.
2. The article does not have a section conclusion to summarize the study. Maybe try to write a few sentences summarizing your research.
Best regards,
Author Response
Dear reviewer 1.
Thank you for your review. We have considered your concerns and were able to implement the proposed changes. For us this was a fruitful process that improved our manuscript.
I only have a few minor suggestions for you to consider in the revision.
- Section 2. Materials and Methods; 2.1. Data and data preparation; lines 113-116, Table 1., and in some parts in Results and Discussion.
In the model, you used climatic data on the current conditions, which you define with relatively little accuracy; e.g "current climate; variables including current times ". I guess that these are data from the reference period 1981-2010 (such data is in CHELSA). This should be precisely specified in the article. Please add in subsection 2.1 information about the current climate data. (e.g. average values from the period .......). As your article shows, the climatic input data has a great influence on the final result of the model, hence information on this is necessary.
RESPONSE: In our dataset the time slice was 1979 to 2013. We specified that in the Text.
- The article does not have a section conclusion to summarize the study. Maybe try to write a few sentences summarizing your research.
RESPONSE: We included a short conclusion paragraph